

PeerJ Hubs
Published on behalf of

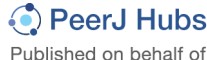

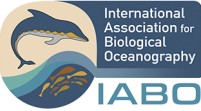

# Using transplantation to restore seagrass meadows in a protected South African lagoon

Katie M. Watson[1], Deena Pillay[2] and Sophie von der Heyden[1,3]

[1] Department of Botany and Zoology, University of Stellenbosch, Stellenbosch, South Africa
[2] Marine and Antarctic Centre for Innovation and Sustainability, Department of Biological Sciences, University of Cape Town, Cape Town, South Africa
[3] School of Climate Studies, University of Stellenbosch, Stellenbosch, South Africa

Corresponding author
Sophie von der Heyden,
svdh@sun.ac.za

## ABSTRACT

**Background:** Seagrass meadows provide valuable ecosystem services but are threatened by global change pressures, and there is growing concern that the functions seagrasses perform within an ecosystem will be reduced or lost without intervention. Restoration has become an integral part of coastal management in response to major seagrass declines, but is often context dependent, requiring an assessment of methods to maximise restoration success. Here we investigate the use of different restoration strategies for the endangered *Zostera capensis* in South Africa.

**Methods:** We assessed restoration feasibility by establishing seagrass transplant plots based on different transplant source materials (diameter (ø) 10 cm cores and anchored individual shoots), planting patterns (line, dense, bullseye) and planting site (upper, upper-mid and mid-intertidal zones). Monitoring of area cover, shoot length, and macrofaunal diversity was conducted over 18 months.

**Results:** Mixed model analysis showed distinct effects of transplant material used, planting pattern and site on transplant survival and area cover. Significant declines in seagrass cover across all treatments was recorded post-transplantation (2 months), followed by a period of recovery. Of the transplants that persisted after 18 months of monitoring (~58% plots survived across all treatments), seagrass area cover increased (~112%) and in some cases expanded by over >400% cover, depending on type of transplant material, planting arrangement and site. Higher bioturbator pressure from sandprawns (*Kraussillichirus kraussi*) significantly reduced transplant survival and area cover. Transplant plots were colonised by invertebrates, including seagrass specialists, such as South Africa's most endangered marine invertebrate, the false-eelgrass limpet (*Siphonaria compressa*). For future seagrass restoration projects, transplanting cores was deemed the best method, showing higher long-term persistence and cover, however this approach is also resource intensive with potentially negative impacts on donor meadows at larger scales. There is a clear need for further research to address *Z. capensis* restoration scalability and improve long-term transplant persistence.

## INTRODUCTION

Seagrasses play fundamental roles in coastal ecology and provide an array of ecosystem services, such as enhancing coastal resilience and contributing to climate stability (*Himes-Cornell, Pendleton & Atiyah, 2018*; *Macreadie et al., 2021*). Despite increasing global efforts to conserve seagrasses, most remain under significant pressure, with accelerating rates of meadow loss and degradation (*Waycott et al., 2009*; *Dunic et al., 2021*). As such, management strategies are urgently needed to facilitate restoration and expansion of meadows (*Unsworth et al., 2019b*). A wave of international restoration targets and policies (*e.g.*, UN Strategic Plan for Biodiversity; *Convention on Biological Diversity (CBD), 2014*) have initiated global ecosystem restoration action (*Waltham et al., 2020*; *Nicholson et al., 2021*; *Buelow et al., 2022*), such as the UN Decade on Ecosystem Restoration (2021–2030). Numerous efforts to restore seagrass meadows have already been undertaken with variable outcomes, but many trials have focused on the northern hemisphere, specifically on *Zostera marina* (*van Katwijk et al., 2016*). Seagrass rehabilitation and restoration remains challenging, with a need for rapid methodological and technological advancements that can be implemented across different species and sites (*Bayraktarov et al., 2016*; *Unsworth et al., 2019b*).

In South Africa, widespread and cumulative global change pressure has resulted in population declines and localised extinctions of the endangered *Zostera capensis* (*Adams, 2016*; *Watson et al., 2023*). Distributed across estuaries on the South African coast, *Z. capensis* is the dominant temperate seagrass species, but populations are likely isolated with low levels of gene flow between them (*Phair et al., 2019*; *Jackson, 2022*). Declines in habitat quality, in conjunction with natural dynamics in estuarine conditions have resulted in fluctuations in meadow cover throughout its range (*Pillay et al., 2010*; *Bandeira et al., 2014*; *Adams, 2016*) with declines in *Z. capensis* cover estimated at 8.3 ha yr$^{-1}$ (*Raw et al., 2023*). This has also negatively impacted genomic diversity of some *Z. capensis* populations, further threatening the long-term resilience of seagrasses in South Africa (*Phair et al., 2020*). Although fast-growing, *Z. capensis* does not colonise quickly, so management intervention through restoration of anthropogenically-impacted meadows is needed in the region (*Adams, 2016*; *Mokumo, Adams & von der Heyden, 2023*; *Watson et al., 2023*). Presently there is lack of knowledge on how best to implement *Z. capensis* restoration, with one restoration trial in Maputo Bay, Mozambique (*Amone-Mabuto et al., 2022*) and attempts in Klein Brak and Knysna Estuary, South Africa, presenting variable success (*Mokumo, Adams & von der Heyden, 2023*). For example, *Mokumo, Adams & von der Heyden (2023)* reported 100% loss of transplants within three months following a restoration trial, suggesting that environmental variability, as well as local site and seagrass population dynamics are important considerations for regional restoration attempts.

A key challenge for restoration efforts remains selection of the transplant site and donor meadow, and refining transplantation methods to maximise transplant survival (*Fonseca, 2011*; *Cunha et al., 2012*; *Park et al., 2013*; *van Katwijk et al., 2016*). Although *Fonseca, Kenworthy & Thayer (1998)* outlined key selection criteria, applicability of this framework is limited in some cases due to local-scale contextual processes (*Calumpong & Fonseca,*

*2001*; *Lange et al., 2022*; *Mokumo, Adams & von der Heyden, 2023*). Site-specific factors such as water flow, wave action and tidal gradients together with seagrass traits, can have varying impacts on restoration outcomes (*Calumpong & Fonseca, 2001*). Planting across multiple restoration sites can help minimise challenges arising from localised site dynamics, and spatio-temporally spread risks (*van Katwijk et al., 2009*), by increasing the likelihood of seagrass proliferation in suitable growth conditions with low environmental stress (*Short et al., 2002*; *Leschen, Ford & Evans, 2010*; *Wendländer et al., 2019*). However, increasing the number of transplantation sites results in additional costs associated with labour, materials, and importantly, can negatively impact donor meadows (*Bayraktarov et al., 2016*). As such, in small-scale transplantation trials, selection of appropriate restoration and donor sites can improve survival and persistence of transplanted seagrass (*McDonald et al., 2020*). For example, there is considerable phenotypic plasticity across *Zostera* populations (*Peralta et al., 2000*; *Salo, Reusch & Boström, 2015*; *Mvungi & Pillay, 2019*), therefore, source populations may possess traits that can positively or negatively influence transplant performance (*van Katwijk et al., 1998*; *Lewis & Boyer, 2014*; *Novak et al., 2017*), with additional genetic compatibility considerations (*Sinclair et al., 2013*; *Jahnke, Olsen & Procaccini, 2015*; *Pazzaglia et al., 2021*). Abiotic and biotic conditions in transplantation sites can similarly determine the degree of transplant survival and persistence (*Siebert & Branch, 2006*; *van der Heide et al., 2011*; *van Katwijk et al., 2016*; *Ugarelli et al., 2017*).

The importance of local biotic processes, such as interspecific-interactions or the microbiome, for seagrass restoration is gaining traction (*Byers et al., 2006*; *van der Heide et al., 2011*; *Ugarelli et al., 2017*; *Fuggle, Gribben & Marzinelli, 2023*). For example, optimising planting density can promote positive feedbacks resulting from below- and above-ground self-structuring mechanisms (*Bos & van Katwijk, 2007*; *Valdez et al., 2020*). Aggregated spatial configuration of transplants can also benefit seagrass through nutrient sharing between rhizomes in stressful conditions (*Silliman et al., 2015*; *Paulo et al., 2019*), if negative feedbacks from self-shading are avoided (*Ralph et al., 2007*). Seagrasses are typically transplanted as vegetated sediment-intact cores or bare-rooted shoots (with or without anchoring) from healthy donor meadows to a restoration site (*Ganassin & Gibbs, 2008*; *Paulo et al., 2019*; *Curiel et al., 2021*; *Lange et al., 2022*). Transplanting cores is generally recommended as the root and rhizome system remain relatively intact (*Phillips, 1990*; *Fonseca, Kenworthy & Thayer, 1998*) and are translocated with the engineered microbiome (*Fuggle, Gribben & Marzinelli, 2023*). Transplanting bare-rooted shoots is likely to have a lower impact on donor meadows and by selecting the appropriate anchoring material can also promote growth and prevent uprooting by wave action (*van Katwijk et al., 2016*; *Lange et al., 2022*). For example, uncoated metal pegs may add otherwise limiting nutrients (*e.g.*, iron) to the seagrass root zone as the peg corrodes, increasing the absorption capacity for phosphorus and reducing sulphide, thus increasing plant productivity (*Holmer, Duarte & Marbá, 2005*; *Ruiz-Halpern, Macko & Fourqurean, 2008*; *Lange et al., 2022*). Bamboo pegs have also been successfully deployed, facilitating moisture and nutrient retention in intertidal sediment (*Ward et al., 2020*; *Lange et al., 2022*) and through degradation, fertilising sediments to prevent nitrogen and phosphorous

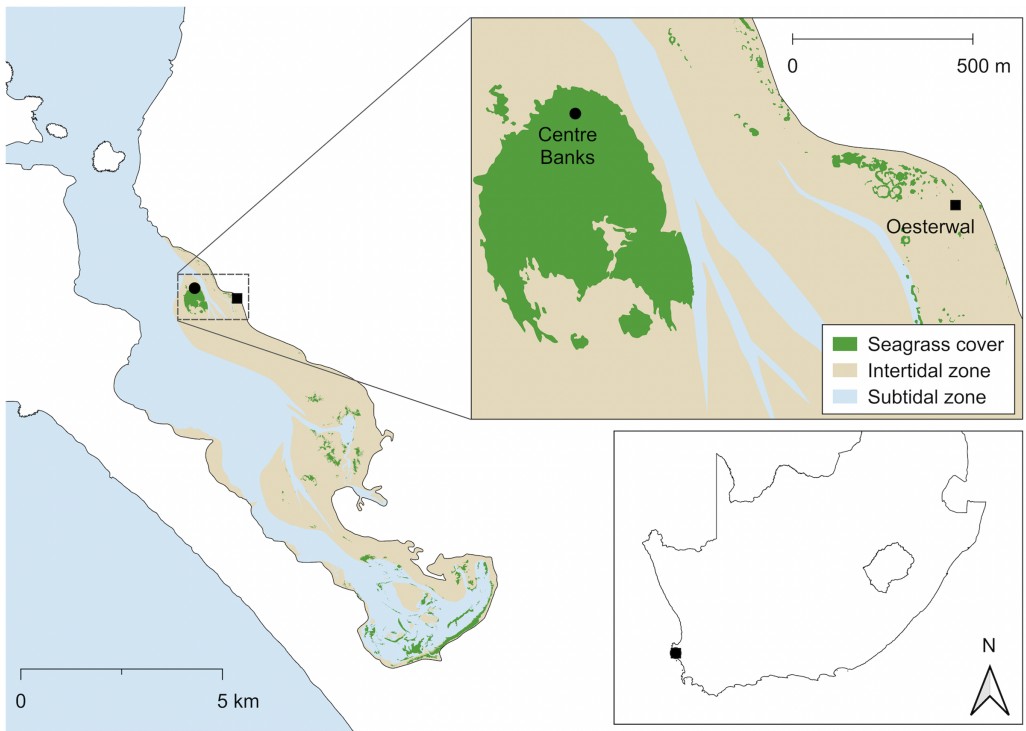

**Figure 1 Location of the donor and restoration sites.** Centre Banks (black circle) served as the donor site and Oesterwal (black square) as the transplant site in Langebaan Lagoon. Inset: Location of Langebaan Lagoon in South Africa (boundary and jurisdiction layers from *DIVA-GIS, 2017*; *QGIS, 2022*)

deficiencies during post-transplant recovery, and releasing silica, aiding growth-promoting bacteria in the root zone (*Ruiz-Halpern, Macko & Fourqurean, 2008*). As such, there are numerous considerations when planning seagrass restoration.

In this study, we investigated the effectiveness of transplanting *Z. capensis* cores and anchored shorts (using both metal pegs and bamboo pegs) across different tidal zones and in three planting patterns, followed by monitoring of seagrass population dynamics for 18 months. Specifically, we aimed to investigate the following parameters as determinants of persistence and cover of transplanted *Z. capensis*: (i) type of transplant donor material (cores and anchored shoots); (ii) the effect of planting pattern; and (iii) the effect of site, as well as the impact of transplanted meadows on seagrass-associated macrofaunal diversity. This foundational research allowed us to identify factors that facilitate seagrass persistence and growth, to support upscaling restoration of *Z. capensis*.

## MATERIALS AND METHODS

### Site description

Langebaan Lagoon (33°08′ S, 18°03′ E) is a marine protected area forming part of the West Coast National Park, covering ~280 km² and stretches over a 38 km coastline (Fig. 1). It is the only estuarine tidal lagoon in South Africa, found within the cool-temperate bioregion (*van Niekerk et al., 2019*). The lagoon is marine dominated and permanently open to the
sea, with no large freshwater inputs (*Day, 1959*) and characterised by intertidal mud and sandflats that support seagrass beds (*Pillay et al., 2010*). Seagrass extent within Langebaan Lagoon has been heavily impacted by human activities, resulting in ~38% loss of seagrass between 1960 and 2007 (*Pillay et al., 2010*), with some sites losing up to 99% of their seagrass cover (*Pillay et al., 2010*).

## Transplantation site selection

Within Langebaan Lagoon, site selection was based on qualitative assessments using aerial photography over the last three decades. Areas showing repeated loss and recolonisation of seagrass, or presence of growth- limiting macroalgae, were avoided (*Hauxwell et al., 2001*) and areas of sandflats with remnants of small stable seagrass patches targeted (*Lange et al., 2022*). Pre-transplant field inspections assessed the environmental conditions of potential transplantation sites, including identifying potential localised human disturbance, any recently emerged seagrass patches, wave action and water depth range.

The donor site (Centre Banks, 33°07′18″ S, 18°02′42″ E) was selected for transplantation due to meadow extent and stability, and proximity to the restoration site (<1 km; Fig. 1). The restoration site (Oesterwal, 33°07′20″ S, 18°03′21″ E) was evaluated in 2020 as meeting the key site selection criteria for restoration (*Fonseca, Kenworthy & Thayer, 1998*; *Calumpong & Fonseca, 2001*) relative to the donor seagrass meadow since it (1) had comparable wave action, water depths and sediment type (<1.0 m mean sea level and sandy sediment; *Compton, 2001*), (2) had a history of supporting seagrass meadows, (3) had sufficient sandflat area lacking *Z. capensis* and macroalgae prior to restoration, (4) had limited human activity, that could lead to disturbance and (5) could support similar quality seagrass habitat (Fig. 1). All works were conducted under the following permits: Department of Forestry, Fisheries and the Environment (DFFE: RES2021-68, RES2022-17) and South African National Parks (SANParks: CRC/2023/017–2020/V1).

## Transplantation method

Three transplant methods were carried out using donor seagrass from Centre Banks: (1) cores (diameter (ø) = 10 cm) with seagrass plants in original intact sediment; three seagrass shoots bundled and anchored with (2) metal pegs (ø 1.6 mm × ø 15 cm uncoated wire bent into a U-shaped stake), and (3) pre-soaked bamboo pegs (ø 1.2 cm × ø 15 cm bamboo cane with a vertical slit to create a V-shape; Fig. S1). Post collection, seagrass cores were placed in ø 12.5 cm plastic pots lined with cotton sheets to keep the sediment intact. Shoots were harvested using a spade and washed in seawater, thus ensuring a minimum of three rhizomal nodes with roots per shoot. During harvest, high density intertidal inner areas of the donor meadow were targeted over a ~400 m$^2$ area to minimise localised disturbance. Donor material was harvested at low tide and transplanted in three intertidal sites (upper, upper-mid and mid, Fig. S2) at Oesterwal (Fig. 2) over a period of six hours per day with volunteer help. To prevent plant stress, cores and shoots were stored in large plastic containers and covered in pre-soaked cotton sheets to prevent desiccation before being planted by hand. For cores, sediment in the transplant plots was dug out using a corer, then seagrass cores were inserted. The planting depth of each core was carefully aligned to the

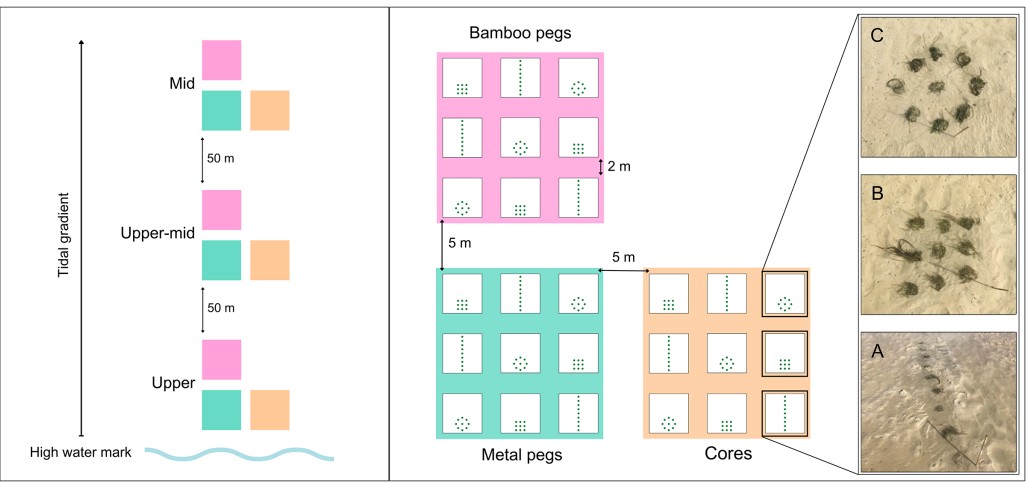

**Figure 2** **Transplantation schematic for restoration sites.** Left: Three transplantation sites across tidal gradient. Right: Transplantation design across each transplant material with planting patterns shown with transplanted cores: (A) line; (B) dense; and (C) bullseye (Figs. S1 and S2). Photo credit: Katie M. Watson.

sediment surface, with displaced sediment redistributed amongst the core and the adjacent sandflat to create a homogeneous elevation of the planting plot (*Paulo et al., 2019*). Shoots were bundled into groups of three per peg, with five pegs per equivalent core (15 shoots/core, from here on also referred to as cores; Fig. S1). The shoots were anchored with pegs and the rhizome was gently pushed 10 cm into the sediment (*Lange et al., 2022*).

Transplantation was conducted in a phased approach in 2021 during the austral winter. Translocation was staggered to spatio-temporally minimise risks during the transplantation phase (*van Katwijk et al., 2009*). Cores were transplanted between 26–28[th] May, shoots anchored with metal pegs were transplanted between 27–30[th] July, and shoots anchored with bamboo pegs were transplanted between 23–26[th] August. Each type of transplant material was planted into spatially discrete transplant plots incorporating nine cores, using three different planting patterns: line, dense and bullseye (Fig. 2). Planting patterns were replicated three times per site, using fully crossed and repeated measures (Fig. 2). To account for differences in water depth, bioturbation, and physical stress (desiccation and wave stress) along the tidal gradient, treatments were replicated at three sites from the upper, upper-mid and mid-intertidal zone, separated by 50 m equidistant intervals (Figs. 2 and S2). A total of 81 transplant plots were planted using 243 cores and 7,290 shoots, with treatment replicates representing ~7.29 m² of transplanted *Z. capensis*.

## Post-transplant monitoring

All transplant plots were left to acclimate overnight before photos were taken ~1.5 m above each plot to obtain baseline seagrass area data using an iPad Pro (1[st] generation, Apple Inc., Cupertino, CA, USA) at low tide the following day. Following the transplants, a regular monitoring protocol was established; at first, monitoring of plots was conducted at low tide weekly for the first eight weeks post-transplant, then fortnightly for the following two months, then monthly, for a total of 18 months following transplantation for each planting

material. Transplant plots were monitored over the following dates cores: 28[th] May 2021–13[th] October 2022, shoots anchored with metal pegs: 30[th] July 2021–15[th] December 2022, and shoots anchored with bamboo pegs: 26[th] August 2021–16[th] January 2023. Photos of each transplant plot were recorded to analyse seagrass area cover ($cm^2$) and sandprawn bioturbator activity (determined by number of excavated *K. kraussi* holes per plot). Measures of epiphytic fouling (% relative cover), three seagrass blade lengths (mm) to determine average canopy height, and abundance of macro-epifaunal species were also taken for each transplant plot.

## Data analysis

Within each transplant plot, area cover ($cm^2$) and excavated *K. kraussi* hole density were analysed in ImageJ (version 1.54d; *Schneider, Rasband & Eliceiri, 2012*). Average canopy height (mm) was determined at each monitoring point by randomly measuring three shoots per plot, from the sediment to leaf tip, and taking the average. To calculate macrofaunal species diversity, the Shannon-Wiener diversity index was calculated with the *vegan* package. The transplant plot survival rate was obtained through the following equation:

$$\frac{\text{(Current no. of seagrass cores)}}{\text{(Initial no. of transplanted seagrass cores)}} \times 100$$

All analyses were carried out in R (version 4.3.0; *R Core Team, 2023*). Generalized linear mixed models (GLMM) were used to examine the effect of the response variables on transplants survival and area cover over time. The response variables: planting material, planting pattern, site, epiphyte coverage (%) and number of sandprawn holes, were included as fixed factors. Transplant plot and monitoring timepoint (number of days since transplantation was used as monitoring was more frequent initially post-transplantation) were included as random effects, to account for taking repeated measurements of transplant plots over time. Normality and homoscedasticity of variance were assessed using the Shapiro-Wilk test and Levene's tests respectively. Where these assumptions were not met, data was transformed (log x + 1) and models refitted. Main effects were assessed, followed by inclusion of interactive effects, then the optimal model structure was determined by selecting the model with the lowest Akaike's Information Criterion (AIC) value. Diversity data were tested for normality, and where deviations found, data was transformed (log x + 1) and model refitted. Main effects were assessed as aforementioned, and model structure selected using the lowest AIC value. The *ggplot2* package was used for graphical presentations.

## RESULTS

### Transplant survival (%)

A total of 243 cores and 7,290 shoots were transplanted across 81 transplant plots, with a mean survival rate of ~58% after 18 months across all treatments. Across all study treatments, all transplant plots survived for a minimum of 31 weeks (full dataset: https://github.com/vonderHeydenLab/Watson_et_al_2023_SeagrassRestoration), after which the
loss of transplant plots differed across transplant source material, planting pattern and site (Fig. 3). There were multiple significant main effects and interactions that impacted transplant survival (Table S1).

Cores outperformed other transplant materials, with a survival rate of ~40% after 18 months across all treatments (Fig. 3). The GLMM showed that cores ($p < 0.001$), and shoots anchored with metal pegs ($p < 0.001$) showed positive, significant effects on survival (Table S1). Biotic interactions significantly affected survival, with lower numbers of sandprawn holes ($p = 0.02$), or lower epiphytic fouling ($p < 0.001$), leading to increased survival (Table S1). The interaction between a higher number of sandprawn holes and transplanted cores ($p < 0.001$), significantly reduced survival (Table S1).

The upper-intertidal site delivered the best outcomes for long-term survival (mean survival ~62% across treatments) for transplant plots, across all planting materials and planting patterns (Fig. 3). Transplant survival was highest in the upper-intertidal site, using cores planted in a bullseye pattern (mean survival ~93%; Fig. 3). The interaction between a lower number of sandprawn holes and transplant plots in the upper-mid ($p < 0.001$), and mid-intertidal sites ($p = 0.01$) significantly increased survival (Table S1).

The use of a bullseye planting pattern, in the upper-mid site and a lower number of sandprawn holes interacted with both cores ($p = 0.05$), and shoots anchored with metal pegs to positively affect survival ($p < 0.001$; Table S1). Similarly, survival significantly increased in interactions between cores ($p = 0.02$), or shoots anchored with metal pegs ($p = 0.01$) planted in a bullseye pattern and lower numbers of sandprawn holes (Table S1). The model also showed a significant increase in survival between the interaction of shoots anchored with metal pegs, planted in the bullseye pattern, in the mid-intertidal site, with lower numbers of sandprawn holes ($p = 0.01$; Table S1). In the GLMM intercept, comprising of shoots anchored with bamboo pegs planted in a line pattern in the upper-intertidal planting site, was significant ($p < 0.001$), with negative impacts on survival (Table S1).

## Transplant area (cm$^2$)

In surviving transplant plots, area cover increased on average by ~126% (mean across all treatments), with some plots expanding in area by over ~400% from across 18 months of monitoring (Fig. 4A). The GLMM showed using cores ($p < 0.001$), or shoots anchored with metal pegs ($p < 0.001$) had positive effects on area cover (Table S2). Conversely, area cover decreased when cores ($p < 0.001$) or shoots anchored with metal pegs interacted with higher numbers of sandprawn holes ($p = 0.01$; Table S2). Under the combination of using cores planted in a bullseye pattern in the upper-intertidal site, area cover increased by ~394% (mean across replicates; Fig. 4). The upper-intertidal site showed an average increase in area cover by ~191% (mean across all treatments and replicates; Fig. 4). Area cover was found to be negatively affected by interactions between cores transplanted into the upper-mid ($p = 0.04$), or mid-intertidal ($p < 0.001$) sites, or with higher numbers of sandprawn holes ($p < 0.001$; Table S2). Seagrass area cover increased when transplanted cores interacted with lower number of sandprawn holes in combination with either the

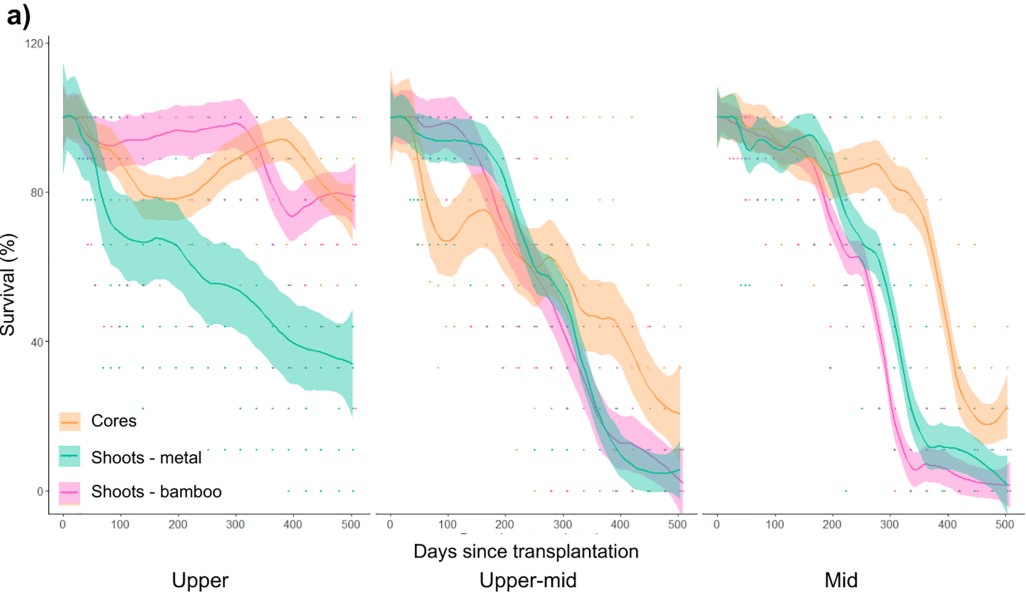

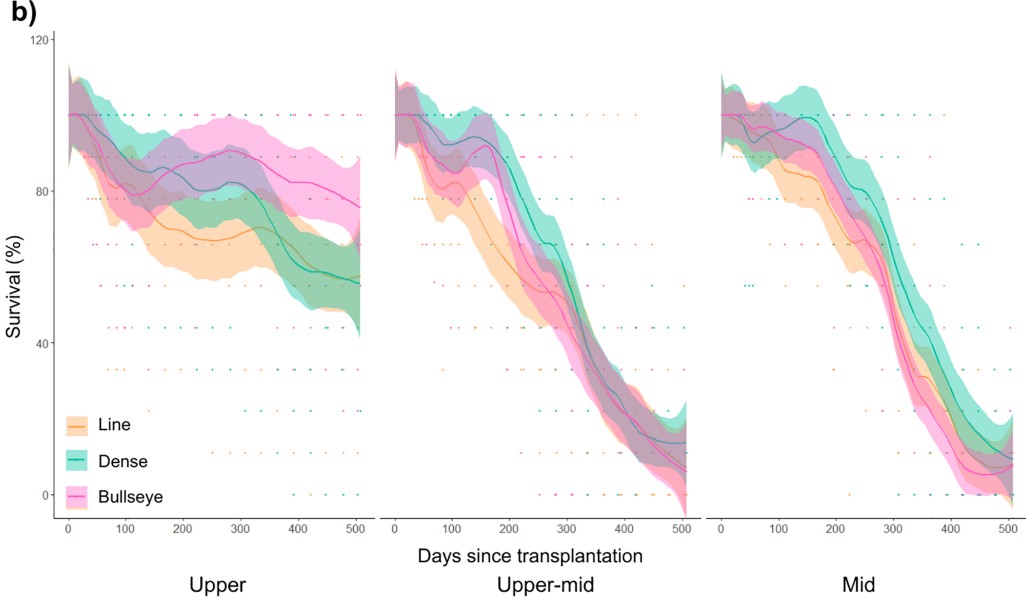

**Figure 3 The effect of transplant material or planting pattern on seagrass area cover (cm²) between sites over 18 months of monitoring.** (A) The effect of transplant material (cores and shoots anchored with bamboo or metal pegs) on seagrass survival (%) across the upper, upper-mid and mid-intertidal transplant sites. (B) The effect of planting pattern (line, dense and bullseye) on seagrass survival (%) across the upper, upper-mid and mid-intertidal transplant sites. Data is represented as mean values across treatment replicates with standard error (± SE).

upper-mid ($p = 0.03$) or mid-intertidal ($p < 0.001$) sites (Table S2). The GLMM intercept, made by shoots anchored with bamboo pegs planted in a line pattern in the upper planting site, was significant ($p < 0.001$), with negative impacts on area cover (Table S2).

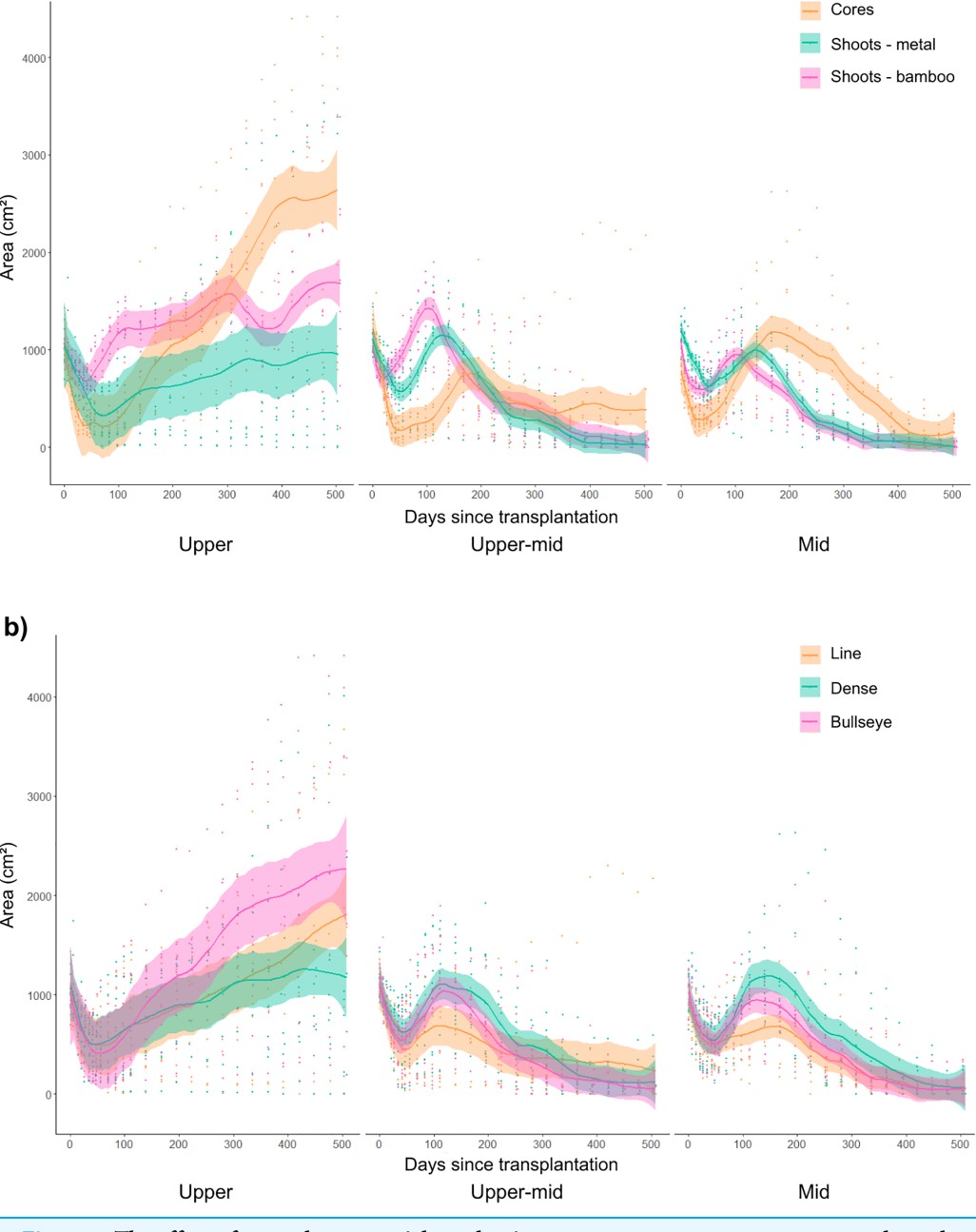

**Figure 4 The effect of transplant material or planting pattern on average seagrass transplant plot area cover (cm$^2$) between intertidal sites over 18 months of monitoring.** (A) The effect of transplant material (cores and shoots anchored with bamboo or metal pegs) on seagrass area cover (cm$^2$) across the upper, upper-mid and mid-intertidal transplant sites (data collated for replicates across different planting patterns). (B) The effect of planting pattern (line, dense and bullseye) on seagrass area cover (cm$^2$) across the upper, upper-mid and mid-intertidal transplant sites (data collated for replicates across different planting materials). Data is represented as mean values across treatment replicates with standard error (± SE).

Transplant area cover increased in transplant plots planted in a bullseye planting patterns, with ~147% (mean across replicates) average increase in seagrass cover after 18 months (Fig. 4B). However, cores planted in a bullseye pattern, with high numbers of sandprawn holes caused negative effects on area cover ($p = 0.01$; Table S2). With a lower number of sandprawn holes this combination of interacting factors (cores, planted in a bullseye pattern) positively affected area cover in both the upper-mid ($p < 0.001$), or mid-intertidal ($p = 0.03$) sites (Table S2). Higher numbers of sandprawn holes ($p < 0.001$), and higher epiphyte cover (%; $p < 0.001$) also had negative independent effects on area cover (Table S2).

All transplant materials were found to undergo a post-transplantation stress phase where seagrass cover initially decreased at 2 months, and then steadily recovered, as shown in Figs. 3 and S3. This pattern was observed across treatments, with more significant recovery in area cover found in transplanted cores, planted in denser planting arrangement on the upper shoreline (Fig. 3). Over time, transplant plots in the upper-mid and mid-intertidal sites steadily declined after signs of recovery, irrespective of transplant material, planting pattern or site (Fig. 3).

Across the monitoring period, 14 macro-epifaunal species were recorded within the transplant plots, including the endangered false-eelgrass limpet (*Siphonaria compressa*). The GLMM indicated that species diversity was positively affected by transplanted cores ($p < 0.001$), but negatively affected by shoots anchored with metal ($p < 0.001$) and bamboo pegs ($p < 0.001$; Table S3). Transplanted cores had the highest Shannon diversity index score, with diversity indices driven by the abundance of sandy anemones (*Bunodactis reynaudi*; full dataset: https://github.com/vonderHeydenLab/Watson_et_al_2023_SeagrassRestoration).

The authors acknowledge that although recovery of the donor meadow was not explicitly monitored from the study outset, a once-off monitoring visit 18 months post-transplantation (October 2022) found no evidence of visible holes or long-term meadow scarring. This suggests that the donor meadow had regenerated across the harvest area, and that sympathetic study design had facilitated donor meadow recovery without intervention.

## DISCUSSION

Our work describes the first trial showing signs of successful restoration using transplantation of *Z. capensis* meadows in South Africa, by transplanting different types of donor material (cores and anchored shoots), in various planting patterns across different sites. Our results highlight that for Langebaan Lagoon, transplanted cores, in a bullseye planting pattern in the upper-intertidal shoreline had the best survival and increase in area cover. Under this combination of treatments, transplantation resulted in established seagrass patches that persisted after 18 months of monitoring and increased in area cover (average ~394% increase). Our work offers valuable insights into the potential for transplantation to be employed nationally by building the foundational knowledge needed for cross-site and large-scale seagrass restoration projects in the future.

## Importance of site selection for transplant survival

The significance of site selection in restoration initiatives is well acknowledged by restoration ecologists. This study employed widely used restoration strategies to determine the applicability of current transplantation best practise in a South African context, showing that planting in the upper-intertidal shorelines increases seagrass survival (average ~62% across all treatments) and area cover (average ~191% across all treatments), compared to upper-mid and mid sites. Several studies have demonstrated that identifying sites with suitable growing conditions continues to require a disproportionately high number of transplantation trials (*Leschen, Ford & Evans, 2010*; *Wendländer et al., 2019*; *Lange et al., 2022*). By incorporating best practice guidelines to rigorously inspect field sites prior to transplantation, this likely improved restoration outcomes. For example, targeting areas with a history of seagrass growth, assessing local environmental conditions, and ensuring minimal anthropogenic disturbance all improved our understanding of local site dynamics (*Fonseca, Kenworthy & Thayer, 1998*; *Calumpong & Fonseca, 2001*). Our approach also allowed identification of potential biotic stress factors, including bioturbation and algal growth, with opportunities to integrate this work into habitat suitability modelling (*Erftemeijer et al., 2023*).

## The effect of transplantation material and method on seagrass persistence and expansion

Implementing restoration projects using cores and anchored shoots is typically a trade-off between the desire for sufficiently resilient transplants (*i.e.*, cores containing an engineered microbiome), and reducing donor meadow impacts. Post-transplantation stress is reduced in cores, and the risk of losing the transplanted area due to stochastic disturbance events is also lowered (*van Katwijk et al., 2016*; *Paulo et al., 2019*). In this study, transplanted cores outperformed other transplant materials, showing the highest survival (average ~40% across all treatments) and led to significantly increased area cover (average ~126% across all treatments). This outcome mirrors the results from the global analysis by *van Katwijk et al. (2016)*, with cores showing higher integrated success than transplanted shoots anchored with pegs. Monitoring of the donor meadow after 18 months provided evidence that sensitive removal of donor material does not negatively impact donor meadows in the long-term.

Experimental seagrass transplant trials have suggested that using more compact planting patterns promotes long-term transplant persistence, particularly in isolated transplant plots, as clumping plants mimic dense root mats, promoting self-facilitation *via* sediment stabilisation and nutrient sharing (*Morris & Doak, 2002*; *van Katwijk et al., 2016*; *Temmink et al., 2020*). From our research, more compact patterns (dense and bullseye), showed higher survival, and the bullseye planting pattern significantly increasing area cover (average ~147%). This result may be linked to the more influential factor of site selection, which at a local scale may lead to interactions of spatially isolated *Z. capensis* transplant plots with other ecosystem engineering processes, such as bioturbators. In Langebaan Lagoon, *Z. capensis* has been shown to be spatially excluded by sandprawns (*K. kraussi*) in higher densities, that subsequently increase sediment suspension and

destabilise sediments (*Siebert & Branch, 2005*; *Hanekom & Russell, 2015*). Qualitative field observations also showed bioturbator pressure from Greater flamingos (*Phoenicopterus roseus*) and Lesser flamingos (*Phoeniconaias minor*) which both form donut-shaped depressions due to trampling and their circular filter feeding technique, resulting in sediment resuspension, and increased nutrient flows, causing root destabilisation and enhanced biofilm production (*Gihwala, Pillay & Varughese,* 2017; *El-Hacen et al., 2018*). Bioturbator interactions in combination with (1) small, spatially isolated transplant plots, (2) planted at sites across a tidal gradient where sandprawn density is known to increase towards the lower intertidal zone (*Siebert & Branch, 2006*), likely led to negative density-dependent effects and account for significantly lower survival rates and area cover in the upper-mid and mid-intertidal zones, in comparison to the upper-intertidal zone, in our study.

## Wider impacts of seagrass restoration in Langebaan Lagoon

In this study, re-establishing seagrass patches has a positive impact on seagrass-associated species, including for South Africa's most endangered marine invertebrate, the false-eelgrass limpet (*S. compressa*), which only lives on *Z. capensis* and is endemic to Langebaan Lagoon (*Angel et al., 2006*). Species diversity was higher in transplanted cores (Table S3), likely due to the increased survival and area cover (Fig. 3). In total, 14 macro-epifaunal species were recorded, which is lower than recorded by *Pillay et al. (2010)* at the nearby Klein Oesterwal, likely explained by the increased survey effort across the tidal gradient and the inclusion of infaunal species in *Pillay et al. (2010)*. Although species diversity was not investigated in bare sandflats, previous studies have highlighted that sandflats without seagrass (including through seagrass loss), have reduced richness and abundance (Orth et al., 2006; *Pillay et al., 2010*).

In addition to ecological service provision, this project also provided social benefits. The community impact of this work resulted in SANParks staff and 40 Project SeaStore volunteers contributing >1,600 working hours to assist with transplantation and monitoring, highlighting the project's success in connecting communities with a typically lesser-known ecosystem. Longer-term, this is also likely to promote stewardship of seagrass meadows and aid future conservation efforts (*Unsworth et al., 2019b*).

## Recommendations for future restoration trials in southern Africa

The principal challenge to upscaling *Z. capensis* restoration is the with vastly different environmental conditions found within estuaries across its range, therefore restoration efforts will need to be population and site specific (*Adams, 2016*). For example, attempts in two estuarine systems in South Africa, Klein Brak and the Knysna Estuary, both showed a 100% loss of *Z. capensis* transplants within 3-months (*Mokumo, Adams & von der Heyden, 2023*), despite following a similar approach to the work presented here. Both estuaries have strong freshwater inflows and fluctuations in physicochemical conditions, water levels and turbidity (*Mokumo, Adams & von der Heyden, 2023*), making their environments more variable than the conditions found in Langebaan, with changes in salinity a likely variable in the loss of transplanted cores in Klein Brak. Conversely, a restoration project in a

relatively protected site characterised by sandy and muddy flats in Maputo Bay, Mozambique, also without significant freshwater flow, recorded survival rates of 75% after 12 months for transplanted *Z. capensis* cores (*Amone-Mabuto et al., 2022*). These studies highlight that site conditions and seagrass population dynamics are significant considerations before extending restoration attempts into other South African estuaries.

Another key challenge of working with *Z. capensis* is the lack of abundant seeds, making restoration using seeds unfeasible despite having shown considerable success in other *Zostera* species (*Marion & Orth, 2010*; *Unsworth et al., 2019a*). Work by *Jackson (2022)* discovered low intrapopulation clonality, which suggests sexual reproduction is more prevalent than once thought and a few inflorescences have been documented in several South African estuaries including Berg, Breede and Olifants in the west, and uMhlathuze, Mngazana and Swartkops estuaries on the south-east coast. However, without pursing seed storage (*Yue et al., 2019*) or controlled assisted evolution (*Pazzaglia et al., 2021*), it remains unlikely that the use of seeds for *Z. capensis* restoration will be an immediately viable option. Alternatively, pursuing techniques to reduce impacts on donor meadows, such as *in vitro* seagrass propagation methods (J Stephens, 2021, unpublished data) or growing donor material in seagrass aquariums to increase cover before transplanting back into the field (A Bossert, 2023, unpublished data) could also advance sustainable restoration efforts of dynamic meadows.

Retrospectively, bioturbator disturbance from flamingos was deemed to have more of an impact than anticipated, but was not explicitly monitored during this study. To overcome bioturbator pressure (sandprawns, flamingos and other wading birds), there is an opportunity for future restoration attempts to pursue ecologically sympathetic bioturbator exclusion methods. Novel approaches could also employ bird guano fertilisation to accelerate recovery (*Kenworthy et al., 2018*), or biodegradable establishment structures that mimic dense belowground root mats and suppress sediment mobility (*Temmink et al., 2020*). It is likely that supporting transplants with bioturbator exclusion measures would reduce post-transplantation stress and promote long-term persistence.

## CONCLUSIONS

In summary, our work describes the first trial to show successful restoration of *Z. capensis* meadows in South Africa, highlighting that intertidal site selection, transplantation material used, and to a lesser degree planting pattern, are significant drivers of transplant persistence (Fig. 3) and promoted the establishment and expansion of *Z. capensis* in this pilot study. This research approach creates a solid foundation from which future projects can build upon to reverse the anthropogenically-driven losses of seagrass in Langebaan Lagoon. Further pilot studies are needed to scale our approach across other temperate South African estuaries where *Z. capensis* is found. In the global context, *Z. capensis* is an endangered species that is endemic to southern Africa, and there have been ongoing calls for wider management and restoration of *Z. capensis* across its range (*Pillay et al., 2010*; *Adams, 2016*; *Barnes & Claassens, 2020*; *Amone-Mabuto et al., 2022*; *Mokumo, Adams & von der Heyden, 2023*). With input from regional stakeholders, the conservation of *Z. capensis* requires a concerted combination of adaptive management strategies, including

restoration, to enable practitioners to implement a holistic approach to regional coastal management.

## ACKNOWLEDGEMENTS

The authors are grateful for the support provided by South African National Parks staff from the West Coast National Park for their invaluable assistance in the field, including William Brink, Herschel Blake, Luyanda Mjiyakho and Pierre Nel. The authors thank the numerous Project SeaStore volunteers, without which this work would not have been possible, for their assistance during fieldwork: Kira-Lee Courtaillac, Jamila Janna, Aylisa Joulbert, Amore Malan, Megan Jackson, Courtney Gardiner, Jessica Stephens, Caitlin Ching Sent, Seamus Morgan, Wilna Pieterse, Nina Blom, Jordan Engelbrecht, Engela Smit, Elaine Kritzinger, Bianca Mairs, Andrew Searle, Marlianca Labuschagne, Tsepo Mlanaa, Frederick Mokumo, Simon Berge, Stian Griebenow, Lucy Morshuizen, Alex Nieto, Courtney Fagg, Adon Parker, Zakariyya Kayat, Grant Evans, Rence Jaccbs, Nicholas Scott, Emma Rossouw, Erica Spotswood-Nielson, Darren Jackson, Josh Frank, Stephanie Schoaeman, Katrien Hartzer and Jessie Yuill. We also thank Megan Jackson, Isabella de Beer, Bianca Jooste and Andrew Jackson for their help with image analysis.

### Funding

This research work was supported by funding from the National Research Foundation (NRF) through the Marine and Coastal Research funding instrument (Grant Number 116048). Katie M. Watson was supported through a grant-holder linked doctoral NRF bursary and a scholarship granted by Stellenbosch University's Faculty of Science, Department of Botany and Zoology, and Sophie von der Heyden. The publication fee for this article was supported by IABO Members through the IABO Hub Cooperative Publishing Fund. The funders had no role in study design, data collection and analysis, decision to publish, or preparation of the manuscript.

### Grant Disclosures

The following grant information was disclosed by the authors:
National Research Foundation (NRF): 116048.
Stellenbosch University's Faculty of Science, Department of Botany and Zoology.
IABO Hub Cooperative Publishing Fund.

### Competing Interests

The authors declare that they have no competing interests.

### Author Contributions

- Katie M. Watson conceived and designed the experiments, performed the experiments, analyzed the data, prepared figures and/or tables, authored or reviewed drafts of the article, and approved the final draft.

- Deena Pillay conceived and designed the experiments, performed the experiments, authored or reviewed drafts of the article, and approved the final draft.
- Sophie von der Heyden conceived and designed the experiments, authored or reviewed drafts of the article, and approved the final draft.

### Field Study Permissions

The following information was supplied relating to field study approvals (*i.e.*, approving body and any reference numbers):

Research was conducted under permit granted by South African National Parks (Permit Number CRC/2023/017–2020/V1) and the Department of Fisheries, Forestry and the Environment (RES2021-68, RES2022-17).

### Data Availability

The data is available at GitHub and Zenodo:

- https://github.com/vonderHeydenLab/Watson_et_al_2023_SeagrassRestoration.
- Watson, K. (2023). Transplant survival_Watson et al_PeerJ_*Zostera capensis*_2023.

In PeerJ. Zenodo. https://doi.org/10.5281/zenodo.10065503.

### Supplemental Information

Supplemental information for this article can be found online at http://dx.doi.org/10.7717/peerj.16500#supplemental-information.

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
