# Peer review of "Using transplantation to restore seagrass meadows in a protected South African lagoon"

_PeerJ, doi:10.7717/peerj.16500_

## Round 0.1 · original submission · Major Revisions

Your manuscript has now been evaluated. The reviewers provided detailed comments, and I ask that you consider these carefully when revising the manuscript.

When submitting the revised version of your manuscript, please state in your cover letter point-by-point which changes you have made in response to the reviews and where and why you have refused to follow a particular suggestion. The accordance between the changes and the reviewers' requests should be sufficiently transparent.

Reviewer 1 ·

Basic reporting

Using transplantation to restore seagrass meadows in a protected South African lagoon (#89700)

The manuscript is straightforward and describes results of transplantation experiment to restore seagrass meadows in South Africa. The results provide very important baseline to maximize restoration success for future restoration projects. . I think it is worth publication in Peer-J after major revision.

As the main shortcoming of the manuscript, I would like to point out the lack of clarity between text and tables as well as between Results and Discussion, which confuses the reader.

The Tables S1-S3 should include all coefficients for entire set of effects and their interactions instead only best-fits. I can not understand why it is written that epiphytes or interactions between cores and the upper-mid or mid-intertidal sites (for example) negatively affected area or survival if coefficients are positive (there are not ‘-‘ before the coefficients). Similarly for many others effects. May be I am not right, but usually minus before the coefficient of linear regression means negative effect. If is so, there are numerous contradictions between text and Tables. Please clarify.

Several facts are postulated without explanation. For example, in Results “Across all study treatments, all transplant plots survived for a minimum of 31 week (full dataset: https://github.com/vonderHeydenLab/Watson_et_al_2023_SeagrassRestoration), after which the loss of transplant plots differed across transplant source material, planting pattern and site. Using metal or bamboo pegs as an anchoring method for shoots, did not prove to be as simple, quick or efficient as transplanting cores”. If you postulate this you should to provide comparative data on % of survival. Please check up throughout the text and remove all unexplained postulates or provide evidence supporting your words.

It should be explained what means the values of survival (~40%, ~62% and other). Is it average values or what? If it average, how they were calculated?

It is also very important to guide the reader throughout the manuscript. For example, in the Discussion section, readers will learn that « Species diversity was higher in transplanted cores, planting in bullseye planting patterns in the upper intertidal site” (Lines 461-462). However, the information about number of species and species diversity in different intertidal zones as well as across planting pattern is lacked in the Results or Tables. Similarly, in Discussion it is mentioned that “survival was significantly lower in both shoots anchored with metal (~13 %) and bamboo pegs (~27 %)” than in cores, although such data are lack in the Results. Authors should to check the text carefully to avoid in Discussion declaration of facts which are not mentioned in the Results section. Also all conventions in Figures should be explained. What means color areas around lines in Fig. 3? Is it SD or SE? What means lines? Average values? And others. In Tables S1–S3, it is better to provide full set of effects in 'Interactions’, instead of mentioning it only in the text.

Several minor comments:

Line 173. Please, make it is clear in the text that donor and recipient sites were located in the same lagoon.

Lines 195–197. Fig. 1. The map caption is difficult to understand. I suggest to change as Location of Centre Banks (black circle) served as the donor site and Oesterwal (black square) as the transplant site in the Langebaan Lagoon. Inset: Location of Langebaan Lagoon in South Africa.

Lines 265-266. Generalized Linear mixed models (GLMM) were used to examine. Also, plese give GLMM in Table captions in full.

Line 282. ~58%. Is it average value? Similarly, please explain this in Lines 288, 289, 292 and throughout the text.

Line 297. significant effects on survival relative to shoots with bamboo pegs (please provide the p-value)

Line 301. significantly affected survival. How? Positively? Negatively? Similarly in lines 305 and 306

Line 303. shoots anchored with metal pegs and lower numbers of sandprawn holes at bullseye planting?

Lines 319-320. Higher numbers of sandprawn holes (p < 0.001), and higher epiphyte cover (%; p < 0.001) also had negative independent effects on area cover. Nevertheless, in the Table S2 coefficient for Sandprawn holes is -4.272 (negative), whereas for epiphyte it is 4.893 (positive). Please clarify.

Table S2. Why SE sometimes by two orders exceed the maen values (Intersept, Material (Cores), Material (metal), Material(Cores):Site(Upper-mid), Material(Cores):Site(Mid). Please explain.

Line 362. It should be mention how macro-epifaunal species were distributed across intertidal zone, and various planting patterns.

Lines 393-394. In contrast, survival was significantly lower in both shoots anchored with metal (~13 %) and bamboo pegs (~27 %). These data lack in the Results

Line 406. Please add point after as bioturbators.

Line 419. I suggest adding the sentence: “These facts can explain lower survival rate and area cover at lower density planting in mid and upper-intertidal zones in compare with upper zone in our experiment.”

Lines 461-462. Species diversity was higher in transplanted cores, planting in bullseye planting patterns in the upper intertidal site. These data is lacked in Results, there is only table with coefficients of regression.

Experimental design

It is adequate

Validity of the findings

no comment

·

Basic reporting

The basic reporting is good but my attachment makes some comments that would improve the manuscript. Overall an impressive piece of work.
these include:
Introduction
Shorten the Introduction it repeats itself through multiple paragraphs.
Lines 88 – 117: these two paragraphs should be combined as information is repeated. I would incorporate the second paragraph into the first paragraph.
Lines 118-147: these two paragraphs should be combined. I would start with the second paragraph and pull sentences out of first paragraph to make the story about the importance of local biotic effects and how methods like spatial configuration and techniques like cores and anchored sprigs can mitigate the local biotic effects
Lines 148-150: Remove the outcome from this paragraph. It is confusing and not an aim. Delete “..conducted the first field trial to show signs of successful restoration of Z. capensis in South Africa, helping to develop foundational principles needed for large scale restoration. Our experiment…”
Materials and Methods
Needs some tightening as suggested in my comments
Lines 221-223: The authors state “Cores were transplanted between 26 . 28th May, shoots anchored with metal pegs were transplanted between 27 . 30th July, and shoots anchored with bamboo pegs were transplanted between 23 . 26th August” This confounds time of planting with methodology so no direct test of methodology for Restoration is appropriate. I understand the difficulty of doing these methods at one time but am then wary of overstating the comparison.
Lines 247-248: Should be deleted as it is not even acceptable English and is confusing.
Lines 247-254: Confusing and I have suggested you remove one sentence as it is explained well in the following section Data Analysis.
Line 265 - 271: I did not understand the design of the Linear mixed models and how the response variables, cores and shoots with bamboo and wire pegs were included in the model, noting my comment above. Nor was there much discussion about covariance in response variables. Note they seem to be fixed but I don’t believe they can be included together in the model as Methods of transplanting are confounded by planting time?
I struggled with the messaging in the results.
Clearly there is a tidal height and methodology signal with higher in the intertidal dramatically better for transplant survival using all methods. The cores also outperformed on survival and growth than the shoot methodologies. Transplanting design (line, dense and bullseye) had a lesser effect but clearly bullseye was most successful over the establishment phase. These messages need to leap out of the results and feature prominently in as first sentences of paragraphs. The present structure is confusing. I would suggest a substantial rewrite of results to give a clearer message.
Overall the figures and tables are scattered between the main body of text and supplementary materials and very difficult to assess. I have some recommendations below:
Table S1 is the linear mixed model outcomes and given its prominence in making interpretations I would prefer it in the body of text.
Table 1: is useful but I would prefer a figure showing the area and %survival on the same figure. Where are the error bars to this data or are these totals of 3 replicates?
Figures 1 and 2 could be combined.
Figure 3 is the main outcome and clearly shows tidal height and method influences success statistically.
Figure 4 was nice but is it the figure you wish to have in the main body of the paper?
Discussion

Firstly I found it incredibly long and recommend shortening it a little as suggested below:

Lines 436-455: I recommend you remove this paragraph as you have not done anything about scaling in this publication.

I would move the paragraphs around a little with the importance of site selection section above the effect of transplantation materials.

Conclusions

These are not the conclusions of your study which were location in relation to tidal height, methodology used and to a lesser degree transplantation pattern drove the establishment of restored Z capensis meadows in this pilot study. e.g. “Clearly with scaling and effort the approach used here could turn back the anthropogenically driven losses of seagrasses in this part of the Estuary. To scale our outcomes to all of temperate South Africa location specific pilot studies are needed. In the global context, Z capensis is threatened and endemic to South Africa and requires a concerted mix of management strategies where restoration is but one.” The lead into the messy politics you have presently as conclusions.

Experimental design

I am concerned that the planting methods are separated by months and there has been a large amount of work on transplant timing in the southern hemisphere showing higher transplant survival in late Autumn versus winter. Lines 221-223: The authors state “Cores were transplanted between 26 . 28th May, shoots anchored with metal pegs were transplanted between 27 . 30th July, and shoots anchored with bamboo pegs were transplanted between 23 . 26th August” This confounds time of planting with methodology so no direct test of methodology for Restoration is appropriate. I understand the difficulty of doing these methods at one time but am wary of overstating the comparison.
The authors dealt with this carefully so I don't believe this issue is enough to stop the acceptance of such a tight and needed study.

Validity of the findings

The findings are valid even with my methodology concern expressed above.

Additional comments

Excellent piece of analytical work on a difficult subject.

---

## Round 0.2 · Major Revisions

Your manuscript has now been evaluated. There is still a request for major changes, before I can accept the manuscript. Please carefully consider the comments before submitting the revised version.

When submitting the revised version of your manuscript, please state in your cover letter point-by-point which changes you have made in response to the reviews. If the accordance between the changes and the reviewers' requests is sufficiently transparent, no further reviewing will be needed.

Reviewer 1 ·

Basic reporting

Authors revised the paper. All reviewer comments were fixed and all queries answered. However, some uncertainties remain that must be resolved before the article is accepted.

Experimental design

no comment

Validity of the findings

no comments

Additional comments

It is still unclear in the results that survival rates are presented as mean values. Please indicate this clearly whenever survival values are stated in the text. It is unclear how the average values were calculated, because it is unclear what data are given in Figs. 3 and 4. Figs. (a) The effect of transplant material (cores and shoots anchored with bamboo or metal pegs) on seagrass area cover (cm2) across the upper, upper-mid and mid intertidal transplant sites. Does the data for cores combine values obtained for cores by planting using different methods (Line, Bullseye, Dense)? Similarly for Metal and Bamboo. (b) The effect of planting pattern (line, dense and bullseye) on seagrass area cover (cm2) across the upper, upper-mid and mid intertidal transplant sites. Does the data for Line combine values obtained for Cores, Metal and Bamboo planting using this method? Similarly for Dense and Bullseye. Please explain. Without this explanation it is difficult to understand the results of the study. Also, I can not see in Fig. 3 the combination of using cores planted in a bullseye pattern in the upper-intertidal site (see line 311-312 “Under the combination of using cores planted in a bullseye pattern in the upper-intertidal site, area cover increased by ~394 % (Fig. 3).”)

There are still in some cases inconsistencies between the table and the presentation of the results. You postulate that “the relationships between coefficients and factors are therefore inverse (i.e., factors with negative coefficients (‘-‘) are positively correlated with survival, and positive coefficient values are negatively correlated with survival)”. Nevertheless, in the text effect of factors with negative coefficients (“–“) in some cases noted as positive (for example Lines 274-276 “The GLMM showed that cores (p < 0.001), and shoots anchored with metal pegs (p < 0.001) showed positive, significant effects on survival) and in other ones as negative (for example Lines 284-285 “The interaction between a higher number of sandprawn holes and transplant plots in the upper-mid (p < 0.001), and mid-intertidal sites (p = 0.01) significantly reduced survival (Table S1)” and Lines 316-317 “Area cover also decreased when transplanted cores interacted with higher number of sandprawn holes in combination with either the upper-mid (p = 0.03) or mid-intertidal (p < 0.001) sites (Table S2)” and Lines 324-327 “However, this combination of interacting factors (cores, planted in a bullseye pattern, and higher numbers of sandprawn holes) negatively affected area cover either in the upper-mid (p < 0.001), or mid-intertidal (p = 0.03) sites (Table S2)”.although in these cases the coefficients for factors are given as negative (“–“), so factors should have positive effect). And vise versa, factors with positive coefficients (that should mean negative effect according to remark in caption “relationship between coefficients and factors are inverse”) in the Table noted as having positive effect in the text (for example Line 278 … higher epiphytic fouling leading to increase survival (coefficient is 8.006, Table S1).

Other contradictions between tables and the text are reporting statistically insignificant effects of the factors as significant (for example Line 280 “The interaction between a lower number of sandprawn holes and …. shoots anchored with metal pegs (p = 0.01), significantly increased survival” whereas in the Table S1 this effect is insignificant (p=0.14)). Similarly, coefficient for bamboo pegs to anchor shoots (see Line 276 “whereas survival significantly decreased with the use of bamboo pegs to anchor shoots (p < 0.001; Table S1)”) is lack in the Table. The bamboo pegs are mentioned in the intercept only (See captions of the Tables).

There are cases of misinterpretation of the results of GLMM. For example “Cores planted in a bullseye pattern, with low numbers of sandprawn holes benefited area cover (p = 0.01; Table S2)” But, addition of one more factor (Site) inverse this effect “However, this combination of interacting factors (cores, planted in a bullseye pattern, and higher numbers of sandprawn holes) negatively affected area cover either in the upper-mid (p < 0.001), or mid-intertidal (p = 0.03) sites (Table S2)”. Although, following the Table in both cases effects are positive (coefficient for factor “Material(Cores):Planting pattern(Bullseye):Sandprawn holes » is 2.795 , that means that cores planting in bullseye at high sandprawn density negatively influences on cover, therefore low sandprawn density have a positive effect. The coefficients for other interactions « Material(Cores):Site(Upper-mid):Planting pattern(Bullseye):Sandprawn holes » and « Material(Cores):Site(Mid):Planting pattern(Bullseye):Sandprawn holes » are negative « –2.935 and -2.226« that also means positive effect).
If it is not clear to me, then it will not be clear to readers either.

Please, explain what T0 and T26 means (Line 307).

Please, indicate in Fig. S3, which photo correspond to T1, T10 and T20.

---

## Round 0.3 · accepted · Accept

In the revised version the authors took into consideration all comments and remarks. I recommend accepting the manuscript for publication in PeerJ.

Reviewer 1 ·

Basic reporting

The authors corrected the manuscript and took into account all comments. Now the Manuscript can be accepted for publication

Experimental design

no comment

Validity of the findings

no comment

Additional comments

no comment